# Extracellular Lipids in the Lung and Their Role in Pulmonary Fibrosis

**DOI:** 10.3390/cells11071209

**Published:** 2022-04-03

**Authors:** Olivier Burgy, Sabrina Loriod, Guillaume Beltramo, Philippe Bonniaud

**Affiliations:** 1INSERM UMR 1231 LNC, LabEX LipSTIC, Faculty of Medicine and Pharmacy, University of Bourgogne-Franche Comté, 21000 Dijon, France; olivier.burgy@u-bourgogne.fr (O.B.); sabrina_loriod@etu.u-bourgogne.fr (S.L.); guillaume.beltramo@chu-dijon.fr (G.B.); 2Reference Center for Rare Lung Diseases, University Hospital Dijon-Bourgogne, 21000 Dijon, France; 3Department of Pulmonary Medicine and Intensive Care Unit, University Hospital Dijon-Bourgogne, 21000 Dijon, France

**Keywords:** extracellular lipids, lipid metabolism, idiopathic pulmonary fibrosis

## Abstract

Lipids are major actors and regulators of physiological processes within the lung. Initial research has described their critical role in tissue homeostasis and in orchestrating cellular communication to allow respiration. Over the past decades, a growing body of research has also emphasized how lipids and their metabolism may be altered, contributing to the development and progression of chronic lung diseases such as pulmonary fibrosis. In this review, we first describe the current working model of the mechanisms of lung fibrogenesis before introducing lipids and their cellular metabolism. We then summarize the evidence of altered lipid homeostasis during pulmonary fibrosis, focusing on their extracellular forms. Finally, we highlight how lipid targeting may open avenues to develop therapeutic options for patients with lung fibrosis.

## 1. The Physiopathology of Pulmonary Fibrosis

Idiopathic pulmonary fibrosis (IPF) is a severe chronic and progressive disease of the lung parenchyma, causing significant morbidity through worsening dyspnoea, increasing cough, and overall irremediable functional decline [1]. The prognosis of IPF is poor, with a median survival time of less than five years. IPF development is classically viewed as an abnormal alveolar repair induced by an unknown triggering event leading to myofibroblast proliferation and extracellular matrix (ECM) accumulation in the lungs. Currently, only two drugs (pirfenidone and nintedanib) have been approved for the management of IPF. However, neither of these medications are able to reverse or even stop the progression of fibrosis, and so the patient’s lung function declines gradually over time [2,3]. The field is looking towards new avenues to develop therapeutics for patients with IPF.

The physiopathology of fibrosis is still not completely understood, although several factors and processes have been identified in the last few decades [4,5] (see [6] for a comprehensive review of the cellular and molecular mechanisms of IPF). The current working model is based on impaired cellular crosstalk among the different cells present in the lung [5]. Repeated damages to the lung epithelium activate an initial process of wound healing with the activation of epithelial cells, mainly alveolar type (AT)II cells. In this context, and under specific chronic conditions (e.g., aging, tobacco exposure, genetic predispositions), an aberrantly activated population of epithelial cells located in the distal lung emerges [7,8]. Those cells, also called aberrant basaloid cells, express specific keratin fibers (Krt8 and Krt15) and are believed to be stacked into a transitional state in which their healing activities are impaired and in which they transmit altered messages that activate neighboring cells such as fibroblasts [9]. Activated fibroblasts or myofibroblasts are mesenchymal cells with a crucial role in fibrosis since they produce extracellular matrix components, collagen being the main component. In addition, they are regarded as a population of cells with invasiveness and aggressive properties, with the expression of specific proteins such as the α-smooth muscle actin (α-SMA) or the hyaluronan synthases (HAS). Recent advances and single-cell transcriptomic approaches have highlighted the complexity of the fibroblast populations and their ability to promote repair or pathological extracellular matrix deposition [10,11]. The main source of (myo)fibroblasts is resident lung fibroblasts, which undergo activation to produce an altered extracellular matrix. Indeed, many other cell types seem to participate in the accumulation of activated (myo)fibroblasts, as observed in IPF [12]. ATII cells can experience genetic reprogramming, similar to the epithelial-to-mesenchymal transition (EMT) described in cancer, to acquire mesenchymal features and therefore contribute to the pool of activated fibroblasts [13]. Although many labs have described such activated ATII cells, the part of EMT-derived fibroblasts in IPF remains controversial [14]. Later, additional structural cells have been shown to differentiate into myofibroblast-like cells, such as pleural mesothelial cells. During IPF, pleural cells gain motility via the expression of the canonical myofibroblast marker α-SMA and can invade the lung parenchyma [15]. More recently, the plasticity of endothelial cells has been demonstrated. Under fibrotic conditions, endothelial cells in the lung can also undergo reprogramming and acquire mesenchymal properties via a process termed endothelial-to-mesenchymal transition [16]. A deeper understanding of this endothelial-to-mesenchymal reprogramming identifies and confirms potential therapeutic targets such as galectin-3 [17] Next to endothelium, vascular smooth muscle cells are also involved in pulmonary remodeling with the production of type I collagen [18,19]. In addition, a recent and growing body of literature has described the key role of immune cells in orchestrating the cellular interplay between epithelial cells and fibroblasts [20,21,22,23]. Under physiological conditions, immune cells and macrophages appear highly heterogenous with distinct populations, each of them involved in specific processes such as inflammation or matrix interactions [24]. Studies with single-cell RNA sequencing of lung tissue from patients with IPF confirmed the presence of immune cells and notably identified alterations in macrophages and the presence of plasma B cells in IPF [20,21,22,25,26]. Although the exact role of these cells during fibrosis is not fully understood, they are believed to contribute to the local pro-fibrotic milieu responsible for the abnormal activation of both epithelial and mesenchymal cells. In aging, the activation of alveolar macrophages can be driven by components of the microenvironment produced by epithelial cells [27]. The crosstalk between immune cells and structural (epithelial and mesenchymal) cells seems thus to be a double-edged sword.

The activation of these cells within the lung relies mainly on the interaction of these cells with abnormal ECM, which is a major hallmark in chronic pulmonary diseases since it serves as a source of cellular activator molecules [28]. These signals can lead to the (re)activation of developmental signaling during IPF, such as TGF-β1 or Wnt signaling pathways, which have an important role in fibrotic processes [4,29]. TGF-β1 is the canonical pro-fibrotic cytokine, responsible for fibroblast activation, structural cell reprogramming towards mesenchymal-like cells, and modulation of immune cell functions. In rodents, the adenovirus-mediated overexpression of TGF-β1 leads to a severe and progressive fibrotic response [30]. Consequently, approaches to counteract its signaling are an intense area of research [31]. Crosstalk between fibroblasts and epithelial cells in the lung appears to be key in maintaining tissue homeostasis, and this communication involves cytokines and extracellular mediators regulating developmental signaling. In physiological conditions, fibroblasts control the stemness of ATII cells via paracrine Wnt signaling [32]. The disruption of this crosstalk, for example, after a massive epithelial injury, leads to autocrine Wnt signaling in the ATII compartment and subsequent increased activation of these cells. In fibrosis, Wnt signaling is pathologically increased and can over-activate ATII cells. Beyond TGF-β1 and Wnt, other signaling are activated during IPF such as Sonic hedgehog, Notch, and the more recently unraveled Hippo YAP-TAZ pathway. YAP-TAZ are transcriptional cofactors activated by tissue stiffness and are overexpressed during IPF [33]. In fibroblasts, they promote invasiveness, production of ECM components, and subsequent fibrosis [33]. In epithelial cells, they contribute to the ATII to ATI differentiation [34]. Whether perturbation in YAP-TAZ signaling is involved in the inefficient ATII to ATI transition observed in IPF remains unclear.

The (re)activation of developmental signaling is currently seen as the consequence of many pathological phenomena which take place in the IPF lung. Insult of the pulmonary epithelium is believed as a key initiatory step of IPF. Why epithelium can be unable to repair itself is still largely unknown, although many mechanisms are linked to the disease [5]. Aging is an important factor in IPF and can give rise to non-functional and pro-inflammatory senescent cells [35]. Senescence has gained significant interest in the past decade, and strategies to clear senescent cells during IPF are actively tested [36,37,38,39]. Beyond senescence, hypoxia is a major hallmark of IPF, where scar tissue deposition hampers gas exchange. Hypoxia also gains attention as a predictive marker of the disease [40,41,42]. As a consequence, hypoxia leads to an increase in oxidative stress, dysfunction of mitochondria, and an unfolded protein response [43,44]. Altogether, these mechanisms impair the cell’s normal function and consequently cause organ failure.

The lung is an organ that is rich in lipids and active in terms of lipid metabolism. In the distal lung, a significant amount of lipids is found in the extracellular space, for example, forming the alveolar surfactant. Pulmonary surfactant is seminal in facilitating respiration by reducing surface tension during inspiration and preventing the collapse of alveoli. The surfactant is composed of up to 90% lipids, mainly produced by ATII cells and secreted towards the alveolar space [45,46]. Among the different lipids produced in the lung (see below), phospholipids, and particularly phosphatidylcholine (PC), are highly involved in the composition of surfactants [46]. In addition to phospholipids, fatty acids (FA) are important components of the pulmonary lipid content. Their significant heterogeneity (length, unsaturation) and their role as a precursor to producing active lipids explain their numerous roles within the cells. Arachidonic acid (AA) is a perfect example. Vehiculated thanks to many phospholipids, AA is an important precursor of active lipids such as prostaglandins, leukotrienes, or lipoxins. Many lipids are released in the extracellular milieu. This implies the presence of specific activation systems and the expression of lipid receptors and transporters such as ABCA, also involved in surfactant assembly within ATII cells [46]. Cholesterol represents around 10% of the lipid component of lung surfactants, and it has a crucial role in lowering surface tension [46]. Beyond pulmonary surfactants, lipids shape the extracellular milieu with which cells interact. The lung ECM is a highly complex patchwork of fibrous proteins (mainly collagens) and proteoglycans (heparin, hyaluronic acid) as well as other glycosylated proteins such as fibronectin, laminin, and osteopontin (OPN). There are many examples of the intimal link between lipids and ECM. In the liver, OPN, which accumulates in fibrotic ECM, regulates the metabolism of PC and cholesterol by acting on hepatic P450 cytochrome expression [47]. Mice deficient for *Opn* are less prone to develop fibrosis and have decreased PC content [47]. In the lung, *OPN* is up-regulated in PASMCs stimulated with S1P [48]. The up-regulation of *OPN* is crucial in S1P-induced PASMCs proliferation. The activation of PPARγ suppresses the effects of S1P on OPN expression in PASMCs. Several master regulators, such as Lipin-1/SREBP, have been identified in line with these studies showing the regulation of lipid metabolism by ECM [49]. This regulatory crosstalk seems to work both ways as active lipids also drive the production of ECM components. For example, lysophosphate acid, a bioactive lipid, regulates the production of ECM production (collagen, fibronectin) via the transcriptomic activation of Yes-associated proteins (YAP)/PDZ-binding domain (TAC) in trabecular meshwork cells [50].

In pathological conditions, lipid production and metabolism are dysregulated. Furthermore, a growing body of literature identifies the alteration of lipids and their metabolic pathways as a central feature in lung fibrosis [51,52]. During wound healing and organ fibrosis, lipids serve as mediators in various processes such as the activation or resolution of inflammation. Moreover, these lipids can mediate both paracrine and endocrine cell–cell communication.

In this review, we will describe the various lipids present in the lung and explain their metabolic pathways. Then, we will discuss the abnormalities found in lipid composition and metabolism during various pathophysiological processes related to fibrogenesis before addressing how lipids may open therapeutic avenues for lung fibrogenesis and IPF.

## 2. Overview of Pulmonary Lipids and Their Metabolism

Lipids are a heterogenous group of hydrophobic molecules that can be classified into four main classes: (1) glycerides, including phospholipids, (2) fatty acids (FA), (3) non-glyceride lipids including cholesterol and sphingolipids, and (4) complex lipids (lipoproteins).

### 2.1. Phospholipids

Phospholipids constitute a class of amphiphilic molecules composed of a central glycerol cytoskeleton branched with two fatty acids (sn-1 and sn-2) and a polar group termed as “the head”. This group can be choline, serine, ethanolamine, or inositol. As stated above, pulmonary surfactant is essentially composed of lipids and mainly PCs such as the dipalmitoyl-PC [45]. The synthesis of PCs relies on choline, which is trimethylamine with many functions in cell proliferation, differentiation, migration, and apoptosis [53,54,55]. Increased choline uptake is observed in cancer cells in various organs, including lung carcinoma cells or liver cirrhosis [56,57,58]. Several pathways leading to the synthesis of PCs have been described, with the de novo synthesis pathway known as the Kennedy or cytidine-diphosphate (CDP)-choline pathway [59]. Within the lung, choline is imported from the blood towards ATII cells [60,61,62,63]. Intracellular choline first undergoes phosphorylation before transfer to a cytidylyl triphosphate (CTP) to obtain CDP-choline, which in turn is branched into diacylglycerol (DAG) to produce PC. Next to their role as structural lipids, phospholipids, including PCs, also serve as a source of secondary messengers and active mediators, mainly via FA transportation.

### 2.2. Fatty Acids

FAs are classified as saturated or unsaturated and also by the length of the carbon chain, including short-chain FAs (≤6 carbons), medium-chain FAs (6–12 carbons), long-chain FAs (12–22 carbons), and very-long-chain FAs (more than 22 carbons) [64]. Some FA can be generated endogenously from acetyl-coA and acetyl-coA carboxylase (ACACA) to produce malonyl-coA, which is the active donor for FA biosynthesis, which is elongated to generate a carbon chain [65,66]. The first product is released into the cytoplasm as a 4-carbon carboxylic acid (butyric acid, C4:0). Repeated elongation cycles of the butyric acid molecule allow the production of long carbon chains up to palmitic acid (C16:0) or stearic acid (C18:0). ELOVL enzymes are responsible for the elongation of FAs and are classified into seven groups based on their substrates [67,68]. Saturated short carbon chain FAs (e.g., palmitic acid) are substrates for ELOVL6. In parallel, saturated long and very long carbon chain FAs (e.g., stearic acid) are substrates for ELOVL1 [68,69], while ELOVL3 and ELOVL7 elongate saturated and unsaturated long carbon chains. ELOVL2 and five elongated unsaturated carbon chains such as arachidonic acid (C20:4, AA) [70]. 

FA can be released from phospholipids via the Lands’ cycle, where phospholipids are converted into lysophospholipids (reaction accompanied by the release of the sn-2 FA) under the action of the A2 phospholipase [58,71]. Lysophospholipids can, in turn, be used by lysophosphatidylcholine acyltransferase (LPCAT) enzymes to branch another FA in the sn-2 position and then form another complete phospholipid, or they can be processed to generate active extracellular lipids. Autotaxin is a secreted enzyme with lysophospholipase D (LPD) activity. It hydrolyzes extracellular lysophospholipids such as lysophosphatidylcholine (LPC) to obtain lysophosphate acid (LPA), a known active lipid mediator [72,73,74]. Indeed, extracellular LPA will bind and activate G protein-coupled transmembrane receptors. Six receptors, LPA1 to LPA6, have been described so far [72]. Although circulating LPA levels correlate with autotaxin concentrations, LPA can be generated after hydrolysis of phosphatidic acid (PA) by phospholipases A1 and A2. This reaction releases the FA in the sn-1 and sn-2 positions of the PA to produce LPA [51,72].

Among FAs, arachidonic acid (AA) is a major inflammatory mediator seeing as it can serve as a precursor to the generation of eicosanoids, including prostaglandin, leukotriene, and lipoxins [75,76]. These active lipids control tissue inflammation. The bioavailability of AA is major, and AA is the main FA located in the sn-2 position of phospholipids found linked to the membranes [49]. Within cells, hormonal, physical, or chemical stimuli can activate phospholipase A2, which is responsible for the hydrolysis of the sn-2 FA on phospholipids, leading to the release of AA. Then, free AA is metabolized by prostaglandin synthase isoforms COX-1 and COX-2. COX-1 is the main isoform expressed constitutively in cells, while the isoform COX-2 is activated during inflammation. COX enzymes produce prostaglandin H2 (PGH2), which serve as a precursor to producing other prostaglandins [77]. Alternatively, AA can also be converted into 5-hydroperoxyeicosatetraenoic (HPETE) acid by lipoxygenase, mainly the 5-lipoxygenase (5-LOX, also known as ALOX) to produce leukotrienes [78]. The lipoxins are pro-resolving lipids that play a vital role in reducing excessive tissue injury and chronic inflammation [79]. Lipoxins are synthesized by two pathways from AA and involve different lipoxygenases (5-, 15- or 12- LOX). The first pathway is the conversion of leukotriene A4 into lipoxin A4 and B4 by 12-LOX. In the second pathway, lipoxin can be obtained by the conversion of AA into 15-HPETE by 15-LOX, successively by the conversion of 15-HPETE into 15-OH-leukotriene A4 to produce lipoxin A4 and B4. 

The extracellular localization of lipids implies a system of recognition through specific surface receptors. ABCA3 is a lipid transporter with an ATP binding cassette believed to play a key role in the homeostasis of pulmonary surfactants [80]. CD36 is a transmembrane glycoprotein also known as FA translocase or scavenger receptor class B2 [81]. CD36 is localized in cellular lipid rafts. The FA moves through the bilayer membrane from the outer to the inner leaflet by a flip-flop process in order to be metabolized. CD36 facilitates the transport of long carbon chain FAs such as docosahexaenoic acid (DHA, C22:6) and eicosapentaenoic acid (EPA, C20:5) through the plasma membrane [49]. CD36 is expressed in various cells, including breast and eye epithelial cells, endothelial cells, enterocytes, insulin-responsive cells, and hematopoietic cells such as platelets, monocytes, and macrophages [81,82,83]. The expression of CD36 is associated with the regulation of lipid metabolism and innate immunity. CD36 is involved in tissue inflammation, intestinal fat absorption, lipid storage in adipocytes, and diseases such as obesity, Alzheimer’s, or diabetes [84,85]. 

### 2.3. Non-Glyceride Lipids and Lipoproteins

Ceramides are essential constituents of the plasma membrane. They regulate cell signaling, proliferation, differentiation, and apoptosis [86,87]. Ceramides belong to the sphingolipid family, which is based on sphingosine (fatty alcohol containing an ethylenic bond and an 18-carbon chain) linked to a long-chain FA [88,89]. Ceramides can be produced in the endoplasmic reticulum from a palmitoyl CoA molecule or be recycled in the lysosome via the hydrolysis of glycosphingolipids or sphingomyelin by sphingomyelinase. In parallel to ceramides, sphingosine-1-P (S1P) has gained increasing attention. S1P is produced after ceramidases produce sphingosine from ceramides [90]. Sphingosine is then used by sphingosine kinases (SphK1, SphK2) to form S1P. S1P is a class of bioactive lipids acting both as intracellular and extracellular mediators. S1P regulates several cellular processes involved in cell cycles, apoptosis as well as invasion, migration, and resistance to cancer therapy [91,92,93]. S1P can also be used as a biomarker of diseases such as Alzheimer’s [94]. 

Beyond phospholipids, fatty acids, and their derivatives, cholesterol is a major part of the lipid composition of mammals [95]. Cholesterol is an important lipid involved in the regulation of membrane fluidity and serves as a precursor for steroid hormones synthesis. This steroid can be obtained through diet or synthesized de novo in the intestine and mostly in the liver. 3-Hydroxy-3-methylglutaryl(HMG)-CoA reductase has been identified as a key enzyme in de novo cholesterol synthesis [96]. The activity of this enzyme can be inhibited by a class of pharmacological inhibitors called statins. As cholesterol is a seminal compound of membranes throughout the body, a complex system regulates its transport from the liver to the peripheral tissue and back [97]. In brief, low-density lipoproteins (LDL) are loaded in the liver with cholesterol and other lipids (FA) that are mainly transported as triglycerides and phospholipids. LDL then distributes lipids to the peripheral tissue. In parallel to LDL, high-density lipoproteins (HDL) take up lipids, mainly cholesterol, from the peripheral tissue and transport them back to the liver for recycling or elimination. This cycle appears to be highly regulated, and its dysfunction results in atherosclerosis and systemic inflammation [97].

## 3. Dysregulated Lipids and Their Metabolism during Lung Fibrosis

In recent years, altered lipid metabolism has been confirmed in pulmonary fibrosis as well as in other lung-related conditions, including ARDS [45,98]. Evidence of dysregulated expression of proteins linked to lipid metabolism during lung fibrosis have been compiled in Table 1. Unbiased transcriptomic approaches in tissue from patients with IPF identified key alterations in genes involved in lipid metabolism and regulation [20,22,45]. Using single-cell RNA sequencing on pulmonary cells derived from patients with IPF, an enrichment of gene ontology terms linked to lipids has been shown in tissue-resident alveolar macrophages [20]. As mentioned above, ABCA3 is a seminal transporter involved in lipid secretion and thus surfactant homeostasis (Figure 1). In ATII cells, the expression of genes involved in lipid transport such as ABCA3 is decreased during IPF [99]. Among genetic interstitial lung diseases (ILDs), more than 200 mutations have already been described in *ABCA3*, located on chromosome 16. Patients present heterogeneous phenotypes, from lethal neonatal respiratory distress syndrome to childhood and rarely adult interstitial lung disease [100]. The same observation regarding Abca3 has been made in mice subjected to bleomycin, who also displayed decreased intracellular cholesterol, free fatty acids, and triglycerides [101]. Consistent with the link between senescent ATII and IPF, single-cell RNA sequencing approaches in mice also revealed a disturbed lipid regulation by SREBPs in aged ATII, together with increased cholesterol synthesis [102]. In fibroblasts, the deregulation of lipid metabolism leads to the accumulation of pathological myofibroblasts [103]. 

Beyond transcriptional studies focusing on the expression of metabolic pathways linked to lipids, a growing body of literature has investigated the alteration of the lipids themselves during fibrosis. It appears that phospholipid classes have a unique distribution between the lung cells, with ATII mainly expressing PG and PC, alveolar macrophages expressing PE, and bronchial epithelial cells expressing PI [110]. The comprehensive profiling of the lung tissue collected on mice subjected to bleomycin highlighted significant differences [111]. In comparison to the control tissue, the fibrotic samples were mainly enriched in PC, PG, and cholesterol ester at D7 post-injury, with PC increasing from D7 to D21. Such lipid quantification was also applied in the bleomycin model together with interventional approaches. In a paper investigating VEGF inhibitors as potent anti-fibrotic in the bleomycin model, analysis of the mouse lung revealed increased lipids, such as PC linked to long fatty acids (e.g., PC 36:4) following bleomycin exposure, which returned to normal upon treatment with a VEGF inhibitor [112]. These variations were confirmed by the analysis of genes and proteins involved in the metabolism of these specific lipid species. 

In addition to unbiased lipidomics, several studies have highlighted the accumulation of specific FAs, such as palmitic acid, during IPF [113]. FA modification is a central event in lipid biogenesis. One possible modification is elongation, which determines lipid function and metabolic activity. The role of ELOVL6 in pulmonary fibrosis has been of particular interest because this enzyme is responsible for the conversion of palmitate (C16:0) into stearate (C18:0). ELOVL6 is significantly downregulated in the lungs of patients with IPF and in mice challenged with bleomycin [113]. In addition, stearic acid inhibits the activation of fibroblasts with reduced TGF-β1-activated Smad signaling as well as ECM and collagen production [114]. In parallel, Elovl6-deficient mice exhibit an altered FA composition in the lung accompanied by a more severe fibrotic response upon bleomycin exposure compared with wild-type littermates. In alveolar epithelial cells, the accumulation of palmitic acid results in oxidative stress, subsequent TGF-β1 production, and apoptosis (Figure 1). Consistently, it has been reported that a palmitic acid-enriched high-fat diet increases mortality in mice subjected to bleomycin [113]. In this model, the increased stress of the endoplasmic reticulum in alveolar epithelial cells has been observed, which was dependent on the presence of the lipid receptor CD36. Nevertheless, the association between increased levels of palmitic acid following a high-fat diet and increased mortality after bleomycin remains not completely understood since fatty diets are also linked to systemic inflammation.

Cellular stress is a common phenomenon observed during IPF, mainly in aberrantly activated ATII cells [5]. Interestingly, stress such as ER or mitochondrial stress is also linked to lipid metabolism disorders. Impaired mitochondria is a major hallmark of aberrantly activated ATII during IPF. Mice harboring deletions for mitofusin proteins, orchestrating mitochondria fusion and homeostasis, exhibit increased fibrosis after bleomycin exposure compared with wild-type mice [115]. Mechanistically, mitofusin inhibition hampers phospholipids, particularly PSs in ATII cells, showing the intimate relationship between cellular stress and lipid metabolism. In ATII cells, the stress of the endoplasmic reticulum that is induced during fibrosis promotes lipid production that is dependent on the stearoyl-coenzyme A desaturase 1 enzyme [116]. In those cells, lipid production is necessary to resolve endoplasmic reticulum stress. The pharmacological inhibition of this desaturase exacerbates ER stress in epithelial cells and then potentializes fibrosis.

Cholesterol and related vesicular transport systems are major bioactive lipid species that are dysregulated in many chronic lung diseases, including COPD or asthma [117,118]. The link between cholesterol and fibrosis is becoming progressively clearer. The development of high-fat diet-induced hypercholesterolemia in ApoE null mice leads to systemic inflammation and further lipid accumulation within the lung, causing subsequent fibrosis [119]. In vitro, the cholesterol derivatives 27- or 25-hydroxycholesterol induce α-SMA and type I collagen expression in mesenchymal cells (Figure 1) [117,118]. In both studies, blockade of the TGF-β signaling abolishes the activation properties of these derivatives. In alveolar epithelial cells, the addition of HDL enhances proliferation and migration properties with the activation of AKT and ERK signaling pathways [120]. It should be noted that HDL is a well-known system for the transport of cholesterol from the peripheral tissues back to the liver.

## 4. Extracellular Lipids as Important Regulators of Fibrosis Progression 

Apart from its crucial role in regulating the surface tension of the alveolar walls, surfactant lipids also manage the interplay between ATII cells and the local immune system. For example, specific PC and P-Glycerol entities are capable of modulating alveolar macrophage polarization and function (Figure 2) [121,122]. Surfactant homeostasis and lipid composition are often disturbed in pulmonary diseases [123].

Lipid profiling applied to bronchoalveolar lavage fluid (BALF) in experimental rodent models has uncovered significant alterations and the accumulation of lipids in the extracellular space within the lung [111]. These lipids include a large range of classes, from phospholipids to eicosanoids. In parallel, the role of oxidized lipids has emerged in many chronic pulmonary diseases such as asthma or lung fibrosis [101,124]. Upon bleomycin exposure, secreted oxidized PC can be measured in the BALF of mice [101]. Consequently, lipid-loaded foam cells accumulate in the lungs of those mice, mainly in proximity to ATII cells. The presence of these oxidized lipids seems to be crucial for fibrosis development, seeing as oxidized PCs mitigated macrophage polarization toward an M2-phenotype (Figure 2). Intra-tracheal instillation of oxidized PCs in mice induces a severe fibrotic response [101]. These data suggest that during pulmonary fibrosis (aberrantly activated), ATII cells secrete altered and oxidized lipids into the lung, which drive a pro-fibrotic, M2-type reprogramming of local macrophages, contributing to fibrogenesis. This highlights the role of lipids in shaping the genetic program of the cells to further promote fibrosis. This epithelial-macrophage crosstalk remains to be described, and the potential role of fibroblasts in this impaired communication warrants further investigation.

Lysophosphatidic acid (LPA) is a major bioactive phospholipid acting via G protein-coupled surface receptors, and it is involved in many illnesses, including cancer and fibrosis [52,125,126]. LPA is increased in wild-type mice subjected to bleomycin compared with non-treated mice [127]. LPA can be generated by autotaxins, which are secreted hydrolases converting LPC into LPA (Figure 2) (see chapter 2, [128]). Autotaxin is overexpressed in human IPF as well as after bleomycin exposure in rodents [105]. Autotaxin knock-out in bronchial epithelial cells or macrophages consistently decreased collagen accumulation in mice subjected to bleomycin. Mice lacking LPA receptor 1 exhibit a rapidly observable default in alveolarization [129]. In alveolar epithelial MLE12 cells, the stabilization of the LPA receptor 1 increases cell migration and ERK signaling (Figure 2) [130]. Furthermore, the administration of an LPA antagonist consistently diminishes bleomycin-induced fibrosis by interfering with the activation of myofibroblasts [127]. LPA1 inhibitors are currently being tested in patients with IPF (see chapter 6).

As mentioned above, arachidonic acid is a master fatty acid thanks to its ability to serve as a precursor for complex lipid mediators such as prostaglandins and leukotrienes. Those so-called eicosanoids seem to be mainly produced by immune cells and appear to be important factors in structural cell activation and thus fibrosis development [52,131]. The production of these mediators seems to be dysregulated during lung fibrosis, mainly resulting in enhanced synthesis of leukotrienes versus prostaglandin E(2), establishing a pro-fibrotic environment (Figure 2) [78]. Whether dysregulated eicosanoid leads to fibrosis development or the other way around seems still unclear. Tissue stiffness, a major hallmark of fibrotic tissues, down-regulates COX-2 expression and therefore diminishes prostaglandin production [108]. In parallel, Cox-2 repression appears to be controlled by a number of mechanisms, including epigenetic and MAP3K8 signaling [132,133].

The role of leukotrienes as contributors to the development of fibrosis has been well described. Mice overexpressing leukotriene C4 synthase had worsened pulmonary fibrosis after bleomycin exposure compared with wild-type mice [134]. Inversely, leukotriene signaling blockade using antagonist of the leukotriene receptor or genetic deletion of the enzyme involved in their production (such as 5-lipoxygenase) attenuates bleomycin-induced fibrosis [135,136,137]. Further, the inhibition of leukotriene production impairs Smad-dependent TGF-β signaling in fibroblasts [138]. Leukotrienes are pro-inflammatory molecules that are part of the altered secretome of senescent cells observed during IPF [36,37]. In IPF, 50% of cells expressing the principal enzyme of the leukotriene synthesis ALOX5 also express the senescence marker p16 [106]. Conditioned media from radiation-induced senescent fibroblasts triggers fibrosis pathways in fibroblasts, and this was abolished upon ALOX5 inhibition [106]. Interestingly, senescent fibroblasts isolated from the lungs of patients with IPF aberrantly produce leukotrienes and no prostaglandins (Figure 2). This disbalance is believed to be one of the mechanisms of fibrosis.

Mice with a genetic deletion of the prostaglandin synthase in hematopoietic cells have increased fibrosis after bleomycin compared with normal mice [139]. This is in line with the protective role of prostaglandins in pulmonary fibrosis, which is not completely understood. In the bleomycin model, prostaglandin E2 supplementation starting at D14 post bleomycin did not impact fibrosis development nor increase fibrosis resolution in the model [140]. However, mice that received prostaglandin E2 before the bleomycin challenge developed less severe fibrosis, while mice lacking the prostaglandin E2 synthase had fibrosis similar to their wild-type littermates [140,141]. This surprising result suggests that prostaglandin may be beneficial during the model development and does not interfere with the mechanisms involved in disease progression. This is consistent with the regulation of prostaglandin production by alveolar epithelial cells. Indeed, extracellular ATP can trigger either prostaglandin or pro-inflammatory IL-6 secretion in those cells depending on the activation or not of ionotropic P2X receptors [142]. Many studies support the idea that prostaglandins have a role in fibroblast biology. Prostaglandins reduce the activation of fibroblasts in myofibroblasts by decreasing their proliferation and inhibiting the production of ECM components [143,144,145,146]. However, the biology behind prostaglandin production and action remains to be fully understood. For instance, the transfer of prostaglandin E2 from epithelial cells or T-lymphocytes co-cultured together with fibroblasts diminishes TGF-β1-triggered activation of the mesenchymal cells [147,148]. Altogether, these studies identify prostaglandins as seminal mediators in inter-cellular crosstalk, showing how alveolar epithelial cells and immune cells direct fibroblast activation. ILD fibroblasts can be resistant to prostaglandin E2 [107]. In these cells, resistance to prostaglandins is explained by the hypermethylation of the prostanoid E2 receptor, which can be restored using a DNA methylation inhibitor [149]. Interestingly, patients with prostaglandin-resistant fibroblasts are also patients with more altered lung function [107]. In parallel, IPF fibroblasts also have nonfunctional COX-2 and therefore lack the ability to produce prostaglandins. Impaired prostaglandin responsiveness also affects the protective role of the plasminogen activation [150,151].

In addition to eicosanoids, other circulating lipids such as S1P have emerged as important players in fibrosis development and progression in the lung and other organs [52,90,152,153,154]. In patients with IPF, S1P is increased in the BALF or in the blood compared with control patients or healthy subjects (Figure 2) [155]. The same observations have been made in rodents exposed to bleomycin [156]. More importantly, the accumulation of S1P in BALF correlates with lung function parameters in IPF. Mechanistically, S1P enhances the reprogramming (EMT) of ATII cells induced by TGF-β1 (Figure 2). This underlines that aberrantly activated ATII cells secrete extracellular bioactive lipids able to enhance their trans-differentiation (EMT). Mice with a genetic deletion of the acid sphingomyelinase, an enzyme required to produce ceramide and thus S1P, have reduced pulmonary collagen after bleomycin compared to control mice [156]. Likewise, the S1P lyase, an enzyme able to degrade S1P, is overexpressed in fibrotic tissues during IPF or after bleomycin in mice [109,157]. The genetic deletion of S1P lyase in mice promotes fibrosis while its overexpression in vitro counteracts TGF-β1-induced cell activation and activates autophagy in fibroblasts. Consistently, the reduced expression of S1PL in PBMCs from patients with IPF correlates with the severity of the disease in those patients [109]. S1P can signal by binding to five surface G protein-coupled receptors, and many studies have investigated what occurs when there is a deficiency of these receptors. The adenovirus-mediated inhibition of S1P receptor 3 in mouse lungs leads to reduced inflammatory cell infiltration, histology change, and collagen accumulation in a model of radiation-induced pulmonary fibrosis [158]. The expression of S1P receptor 3 is regulated by the microRNA-495-3p. In alveolar epithelial cells, the microRNA-495-3p mimic decreases S1P receptor 3 and hampers the activation of those cells [158]. In mice, the genetic deletion of the enzyme involved in S1P generation, sphingosine kinase (SPHK)1, in fibroblasts or alveolar epithelial cells, reduces bleomycin-induced fibrosis [159]. The pharmacological inhibition of SPHK1 reduces fibrosis in vivo. Mechanistically, SPHK1 inhibition counteracts the activation of YAP signaling that is triggered by TGF-β1 or bleomycin exposure in fibroblasts or epithelial cells in vitro [159]. In this study, disturbing SPHK1 expression in endothelial cells had no effect. In parallel, invalidation of the S1P receptor 1 gene in endothelial cells resulted in worsened fibrosis after bleomycin exposure [160]. Compared to controls, S1pr1-/- mice have increased vascular permeability and immune cell influx within the lung as well as coagulation activation. Interestingly, increased S1P/S1PR1 axis in ApoM overexpressing transgenic mice did not reduce experimental fibrosis compared with normal mice [160]. S1pr2 null mice exhibited less inflammation and fibrosis upon bleomycin compared with wild-type mice [161]. The pharmacological inhibition of the S1P receptor 2, using JTE-013, diminishes the activation of alveolar epithelial cells induced in vitro by TGF-β1 [161]. However, the use of a genetically engineered mouse model to track S1pr2 expressing cells shows that this receptor is not limited to structural cells. Alveolar macrophages also express this S1P receptor 2 during fibrosis [162]. The analysis of BALF cells from S1pr2-/- mice subjected to bleomycin showed increased IL-13 signaling [162]. This finding is consistent with the hypothesis that S1P receptor 2 engagement on macrophages promotes fibrosis by polarizing those cells towards a pro-fibrotic M2 phenotype (Figure 2) [162,163].

Cellular communication is key in the onset and development of IPF. Several mechanisms of cell-to-cell crosstalk have been described in the disease, one of them being extracellular vesicles (EVs) [5]. EVs, including exosomes, are membranous vesicles secreted by all cells and act as transporters for molecules (also called cargoes) such as proteins, nucleic acids, or lipids [163]. A growing body of literature shows that EVs are a key component in the pathobiology of IPF. Those vesicles accumulate in the lungs of patients with IPF [164], and they harbor specific cargoes of proteins and microRNA [164,165,166,167,168,169]. Mechanistically, fibrosis-derived EVs activate the mechanism of fibrosis, such as fibroblast proliferation and activation, by controlling developmental signaling such as Wnt or TGF-β1 [164,167]. This seems to be highly dependent on the crosstalk driven by these vesicles. Macrophage-derived EVs appear to have anti-fibrotic properties through the transfer of specific microRNA to epithelial cells [170]. Vesicles secreted by bronchial epithelial cells also transfer microRNA and hamper fibroblast activation and senescence [171]. The characterization of these vesicles, the crosstalk they mediate, and the understanding of how cargos are packed into them could potentially lead to the identification of new therapeutic targets in IPF. EVs carry different molecules, such as lipids, that participate in disease mechanisms. These vesicles accumulate in the BALF of patients with asthma, and significant lipidomic changes, mainly on ceramides and PG, have been observed in EVs from asthmatics compared with healthy controls [172]. EVs isolated from asthmatics transport not only specific lipids but also trigger leukotriene production in bronchial epithelial cells, which demonstrates the close link between EVs and lipid dysregulation during chronic lung diseases [173]. In IPF, EVs also carry a distinct cargo of microRNA and proteins [165,174]. The analysis of the proteome of the IPF-EVs shows the presence of proteins involved in lipid metabolism [165]. Furthermore, prostaglandins are found in EVs from fibroblasts activated with IL-1β (Figure 2) [175]. This finding is consistent with the hypothesis of paracrine signaling in which inflammation-primed fibroblasts secrete EVs to communicate with and limit the activation of the surrounding mesenchymal cells.

## 5. Lipids and Pulmonary Fibrosis, The Hope for Potential Biomarkers

Patient heterogeneity and stratification remain a challenge in IPF. Over the last few years, there has been an emphasis on this aspect in order to better classify patients. More particularly, studies have investigated whether lipid profiling can be used as biomarkers or prognostic tools in pulmonary fibrosis. Lipid transport has been identified as a key component of the proteins identified in the BALF of a subset of patients with pulmonary fibrosis [176]. At the circulating level, the quantification of HDL particles correlates inversely with IPF severity and prognosis (Figure 2) [177]. This supports data obtained from larger cohorts such as in the MESA study, where high levels of HDL were associated with lower fibrosis biomarkers (e.g., MMP7, SP-A) and fibrosis assessed by CT [178]. The mechanism underlying this association remains to be fully elucidated, but it is known that lipids obtained through diet and transported by circulating lipoproteins can influence pulmonary surfactant composition [179,180].

Lipid profiling has been applied to plasma collected from patients with IPF. The analysis of the lipid species found in plasma from IPF patients compared with control patients identified a six-lipid signature, including triglycerides and PCs (Figure 2), that could be used to differentiate samples from the two groups [181]. An in-depth lipidomic analysis showed significantly increased triglycerides and PCs in patients with stable vs. rapidly progressing IPF [182]. In the progressor group, the enriched lipidomic networks were the metabolism of linoleic acid, arachidonic acid, and glycerophospholipids.

Although these data from clinical trials are appealing, they need to be challenged in larger and multi-centric cohorts before being potentially used in clinical practice. The heterogeneity of patients diagnosed with IPF will probably represent a major bottleneck in this quest to use a lipid signature to evaluate prognosis or disease progression. Indeed, IPF is often diagnosed in an aged population that likely brings other conditions (and accompanying medication) on top of the disease itself. In particular, the challenge here would come from patients with lipid-lowering drugs (statins, fibrates). Moreover, additional research is needed to rule out the possibility of increased arachidonic acid metabolism due to inflammation, which could be observed during episodes of acute IPF exacerbation. 

## 6. Lipid-Focused Therapies

Lipids are essential components of cellular metabolism within all cells. Consequently, several approaches focused on controlling cell metabolism, including lipid regulation, have been investigated for their anti-fibrotic properties. Metformin, which is used clinically in diabetes, can reverse pulmonary fibrosis after bleomycin administration [183]. In IPF fibroblasts, metformin promotes lipid accumulation and metabolism with enhanced PPAR-γ [11]. In parallel, reduced TGF-β1-mediated activation and collagen production are observed in these cells after exposure to metformin, consistent with a myofibroblast differentiation towards a lipofibroblast phenotype [11]. In addition to cell metabolism, senescence is a major hallmark of IPF, and senolytic drugs are actively being studied as a therapy for fibrotic disorders [5]. The senolytic drug quercetin has a beneficial effect on bleomycin-induced pulmonary fibrosis in mice [184]. Interestingly, quercetin administration has been associated with a decreased S1P signaling in fibroblasts [184]. Another active area of investigation is S1P signaling. Chemical antagonists of S1P receptor 2 have been tested in mice exposed to bleomycin [185]. In this model, when compared to controls, the S1P receptor 2 inhibition seemed to diminish fibrosis as effectively as the anti-fibrotic drug pirfenidone [186].

As described above, increased levels of HDL are associated with less severe disease in patients with IPF [177], suggesting that cholesterol accumulation may be detrimental. Over the last decades, a growing number of cholesterol-lowering drugs have been developed, statins being one of them. The administration of statins seems to have a beneficial effect on bleomycin-induced lung fibrosis in rats [185]. This observation concurs with retrospective studies assessing the effect of statins in patients with IPF [187,188].

Disturbed lipid and fatty acid levels appear to be a key component of fibrosis. Many studies have therefore tried to correct the levels of lipids that decrease during fibrosis. Supplementation with anti-oxidative lipids such as α-lipoic acid decreases fibrosis induced by bleomycin [189]. The effect of the omega-3 fatty acid DHA has also been tested in pre-clinical models. Compared with controls, mice subjected to bleomycin and receiving the protectin DX (a DHA derivative) display reduced fibrosis with enhanced lung function and a longer life span [190]. Protectin DX’s mechanistic ability to reverse fibrosis has been observed, explaining the anti-fibrotic properties of the compound. In vitro, fibroblasts cultured in the presence of the DHA derivative maresin 1 exhibit fewer activation markers upon TGF-β exposure, in addition to decreased α-SMA expression, Smad signaling, and ERK signaling [191]. In mice exposed to bleomycin, the local administration of DHA during the early inflammation phase inhibits collagen accumulation [192]. DHA is associated with decreased levels of eicosanoid production in this model. 

Eicosanoid imbalance and further leukotriene production over prostaglandin is believed to be a dysregulated mechanism that has an important role in fibrosis progression. Leukotriene inhibitors have already been assessed in pre-clinical models of pulmonary fibrosis and were found to decrease fibrosis in mice when administered starting on D7 post bleomycin exposure [193]. Similarly, agonists of the anti-inflammatory lipoxin receptors inhibit collagen accumulation following bleomycin exposition [194].

LPA is another extracellular bioactive lipid with pro-fibrotic activity. Pharmacological inhibition of LPA or autotaxin, which produces LPA, is an active area of research. In the bleomycin model, inhibitors of autotaxin or LAP receptors protect mice from fibrosis [195]. Thanks to encouraging data from pre-clinical models, autotaxin inhibitors are being tested in patients with IPF [196]. 

Lipids can also serve as extracellular mediators that activate cellular signaling by binding to surface receptors such as fatty acid receptor CD36. Recently, this receptor and its inhibition have gained significant attention in the field of cancer as a potential strategy to limit metastasis [197]. CD36 expression correlates with high EMT and poor prognosis in patients with cancer [198]. Its function in lipid-mediated cell reprogramming (EMT) and immune cell activation has led to the testing of CD36 inhibitors in clinical trials in patients with cancer [199]. In silica-induced lung fibrosis in rats, lentivirus-based CD36 silencing diminished hydroxyproline content and other fibrosis-related markers compared to controls [200]. The potential of CD36 inhibitors, while developed in cancer, remains to be investigated in fibrosis

As discussed above, cellular communication is of major importance in IPF, and part of this communication is mediated through EVs [201,202]. There is growing interest regarding the use of EVs isolated from anti-inflammatory and anti-fibrotic mesenchymal stem cells as a therapy for fibrotic disorders [203,204]. Such strategies have been employed in preclinical models and seem to be efficient in reducing experimental lung fibrosis [203,205,206]. These data exist, but more research is required to fully characterize these vesicles and develop production methods for translational studies.

Nevertheless, numerous lipid-targeting strategies have been tested in patients with IPF in line with these preclinical data. The autotoxin inhibitor GLPG190 was tested in a phase 3 study in IPF but was stopped because the benefit-risk profile no longer supported the continuation of the study (NCT03733444). Moreover, high-affinity LPA1 antagonists were assessed vs placebo in a phase 2 study in patients with IPF and found to significantly reduce FVC decline [207]. They are currently being tested in patients with IPF or progressive fibrotic interstitial lung disease (ClinicalTrials.gov Identifier: NCT04308681). Other lipid-targeting drugs are currently being tested in IPF, such as the PBI-4050 compound, which completed an open-label phase 2 clinical trial in IPF [208,209]. PBI-4050 is known to modulate the activity of G-protein coupled lipid receptors GPR40 and GPR84. This molecule was tested alone or in combination with either pirfenidone or nintedanib. The published results from this trial show no significant change in lung function (%FVC) from baseline to three months in patients with PBI-4050 alone or PBI-4050 + nintedanib [208]. 

## Figures and Tables

**Figure 1 cells-11-01209-f001:**
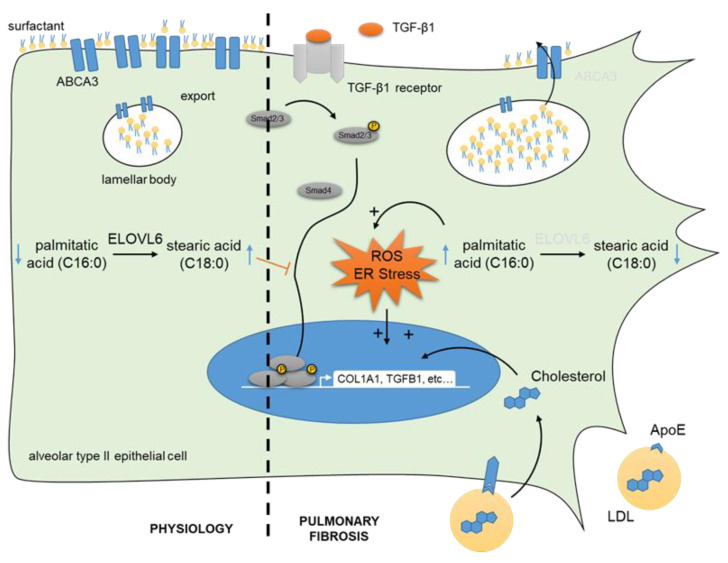
**Altered lipid metabolism during fibrosis contributes to the activation of alveolar epithelial cells.** Under physiological conditions, the elongation enzyme ELOVL6 allows fatty acid elongation and therefore promotes high intracellular stearic acid vs. palmitic acid. Stearic acid interferes with pro-fibrotic TGF-β/Smad signaling (red inhibition arrow). In parallel, ATII cells produce surfactant lipids (mainly PCs) which will be stocked in lamellar bodies and exported in the extracellular milieu in an ABCA3-dependant mechanism. During pulmonary fibrosis, ELOVL6 expression is decreased, favoring the accumulation of palmitic acid. High intracellular palmitic acid induces oxidative and ER stress, thus promoting TGF-β/Smad signaling and cell activation. This is enhanced by the accumulation of cholesterol and its derivates during fibrosis, which activates the expression of collagen and other ECM components. In addition, surfactant lipids accumulate within the cell due to decreased expression of ABCA3 under fibrotic conditions. This also leads to impaired surfactant formation in pulmonary fibrosis.

**Figure 2 cells-11-01209-f002:**
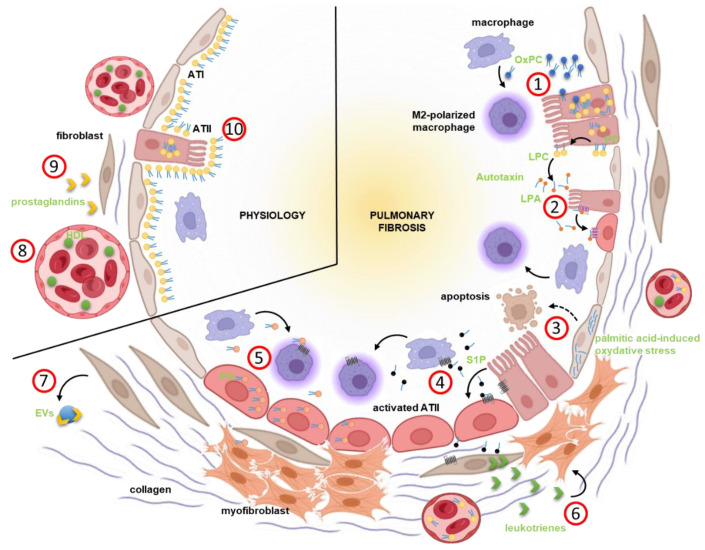
**The role of extracellular lipids in the pathogenesis of pulmonary fibrosis.** During pulmonary fibrosis, (**1**) impaired secretion of the surfactant lipids, mainly phosphatydil-choline (PC) promotes the accumulation of oxidized phospholipids, mainly oxidized-PC (OxPC). (**2**) PC are also a source of lysophosphate acid (LPA) after processing by autotaxin, which is increased during fibrosis. (**3**) The accumulation of palmitic acid in alveolar epithelial cells results in major oxidative stress and ultimately apoptosis. (**4**) Sphingosine-1-phosphate (S1P) accumulates in the extracellular space and promotes the EMT of ATII cells. Further, (**5**) activated ATII cells produce large amount of phosphatydil-glycerol. All these lipids contribute to the polarization of local macrophages towards a profibrotic M2-phenotype or the activation of ATII cells. In addition, S1P has a supplemental role in the activation of aberrant ATII cells. Altogether, cell activation turns fibroblasts into pathological (myo)fibroblasts which (**6**) produce high levels of arachidonic acid derivatives with a high leukotriene to prostaglandin ratio. Of note, (**7**) inflammation-primed fibroblasts also secrete extracellular vesicles carrying prostaglandins. The increase in pulmonary lipids during fibrosis is linked to increased circulating PC. In normal conditions, (**8**) lower levels of blood PC are likely to be found with increased circulating HDL, and (**9**) arachidonic acid metabolism in fibroblasts results in leukotriene overproduction. (**10**) ATII cells produce surfactant-forming lipids (mainly PCs) to participate in lung surfactant homeostasis. Cell types (black) and lipids (light green) are labeled.

**Table 1 cells-11-01209-t001:** Lipid metabolism dysregulation during pulmonary fibrosis.

Proteins Associated with Lipid Metabolism	Regulation in Lung Fibrosis	Evidence	Disease/Model	Cell Type(s)	Reference
ATP Binding Cassette Subfamily A Member 3	down	scRNAseq	patient, bleomycin model	ATII	[99]
Sterol Regulatory Element Binding Transcription Factor 2	up	scRNAseq	bleomycin model	ATII, lipofibroblasts	[102]
Peroxisome Proliferator Activated Receptor Gamma	down	qPCR	IPF	lung tissue	[103]
Elongation of Long Chain Fatty Acids 6	down	qPCR, IHC	patient, bleomycin model	ATII	[104]
Autotaxin	up	IHC, qPCR, ELISA	patient, bleomycin model	hyperplastic bronchiolar and alveolar epithelium, fibroblasts, macrophages	[105]
Arachidonate 5-Lipoxygenase	up	qPCR	bleomycin model	senescent cells	[106]
Leukotriene C4 Synthase	up	qPCR	bleomycin model	senescent cells	[106]
Prostaglandin D2 Synthase	up	qPCR	bleomycin model	senescent cells	[106]
Prostaglandin-Endoperoxide Synthase 2	up	qPCR	bleomycin model	senescent cells	[106]
Prostaglandin E Synthase	up	qPCR	bleomycin model	senescent cells	[106]
Prostaglandin E Receptor 2	down	western blot	IPF	fibroblasts	[107]
Prostaglandin E Synthase	down	IHC	IPF	epithelial cells, fibroblasts	[108]
Sphingosine-1-Phosphate Lyase 1	up	IHC, western blot, qPCR	patient, bleomycin model	fibrotic tissue, fibroblasts, PBMCs	[109]

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
