# Peer review of "Extracellular Lipids in the Lung and Their Role in Pulmonary Fibrosis"

_cells, 2022, doi:10.3390/cells11071209_

Round 1

Reviewer 1 Report

This interesting manuscript gathers and discusses the current literature concerning the impact of lipids on IPF onset and progression.
Other reviews in this field are present (e.g., PMID: 32549377, PMID: 32493486); therefore, the authors should try to differentiate their work for the information's completeness.

With the intent to improve this manuscript, I have some suggestions, which I believe should be incorporated/discussed in the manuscript.

The introduction concerning the cellular and molecular mechanisms involved in IPF etiologic lacks of some important emerging mechanisms such as Epithelial to mesenchymal transition (EMT) and Endothelial to mesenchymal transition (EndMT), and several other (see this recent review for an exhaustive list of mechanisms PMID: 33201251). In this regard, lipids appear involved in modulating IPF-associated EndMt and EMT ( see PMID: 32549377, PMID: 32493486), an aspect that should be discussed in the review. In fact, besides fibroblasts and alveolar epithelial cells, other types of cells such as endothelial and vascular smooth muscle cells appear involved in IPF (see PMID: 33082445, PMID: 33771973, PMID: 30420906)

I believe including the above-mentioned info will improve this "already nice" review

Author Response

We thank the reviewer for the constructive and meaningful comments, which were addressed as follow. We have extended the introduction chapter and now have included more details about the cellular and molecular mechanisms of IPF. According to the amount of additional material, we have changed the title of the first chapter which is now entitled “The physiopathology of pulmonary fibrosis”.

In particular, we have deepened our presentation of the (myo)fibroblast population. We have detailed how several cell types (including epithelial cells, endothelial cells and vascular smooth muscle cells) undergo activation or reprogramming to acquire a myofibroblast-like phenotype. Please see the modifications highlighted in blue at lines 56 to 72. In addition, we have broadened our description of the developmental signaling (TGF-β and YAP-TAZ) involved in IPF and the other mechanisms driving the disease (senescence, hypoxia, ER/oxidative stress). Please see corresponding text at lines 90-97 and 101-120. As advised, we also added the reference from Phan et al., Cell Mol Life Sci. 2021 (PMID: 33201251) to direct the readers to a comprehensive review of the cellular and molecular mechanisms of IPF (see lines 39/40).

Regarding the involvement of lipids and particularly extracellular lipids, we have highlighted in corresponding chapters examples where extracellular lipids impact cell reprogramming. To avoid any confusion, we have specified in the text EMT or EndoMT as needed (e.g. line 480-482). The role of secreted oxidized phosphatidylcholines to polarize macrophages towards a pro-fibrotic M2 phenotype has been better underlined (see lines 386/387). In the same way, we have emphasized the role of S1P as an extracellular bioactive lipid able to drive the EMT of ATII cells during IPF (line 481). This was also pointed out in the legend of figure 2 (please see legend for figure 2, item #4, line 421). Finally, we emphasized the association of the lipid transporter CD36 with EMT in cancer (lines 624/625).

We believe the modifications made here have improved our manuscript and hope our work now represents a comprehensive review on the role of extracellular lipids in pulmonary fibrosis.

Reviewer 2 Report

In the current review, the authors first introduce lipids and their cellular metabolism in the lung. They then summarize the dysregulation of lipids and their metabolism, especially extracellular lipids, during pulmonary fibrosis. Finally, they summarize the Lipid-focused therapies. The overall presentation of the article is good and references are used appropriately. However, some sections do require further summary and elaboration.

  1. Need detailed review of articles on transcriptomic data and summarize the dysregulated genes involved in lipid metabolism and regulation during pulmonary fibrosis. It would be useful to use a table to summarize dysregulated genes associated with lipid metabolism pathways.
  2. Lipids as potential biomarkers in pulmonary fibrosis: Are these lipid biomarkers used to diagnose or evaluate prognosis or disease progression?
  3. Lipid-focused therapies: It would be useful to have a concluding sentence or paragraph talking about the drugs currently being used in lipid-focused therapy.

Author Response

We acknowledge the constructive comments. We have developed the dysregulation of lipid metabolism found in single cell RNA sequencing datasets (lines 279-281). Recent unbiased transcriptomic performed from patient samples uncover much more the upregulation of a signature of genes linked to lipid metabolism and not specific genes. To go further, we have now included a table as suggested to summarize the genes for which a significant regulation of expression has been demonstrated (see table 1, line 295). Thanks to these modifications, we think the readers will have a better overview of the dysregulated genes associated with lipid metabolism pathways during pulmonary fibrosis.

We thank the reviewer for pointing out the need to improve our biomarker-focused chapter. From the literature available, the field most likely moves towards the use of lipids as potential biomarkers to evaluate prognosis and better define intra-IPF endotypes (eg. slow vs. rapid progressors). To avoid any confusion, we have expanded a concluding section in this chapter specifying that all the data reviewed were acquired during clinical trials. Further, we took this opportunity to better outline the major upcoming challenges (eg. validation in larger, multi-centric cohorts) before having lipids as biomarkers in clinical practice (lines 566-571). In addition, we have modified the title of this chapter to emphasize the validation required before the use of lipids into the clinic (line 545). We hope our modifications make the chapter clearer and enlighten the promise of lipids as future potential biomarkers in IPF.

We acknowledge the reviewer’s comment regarding the therapy-focused chapter. To address this point, we have edited the chapter and added an additional paragraph to recapitulate the strategies/drugs being currently tested in clinical trials with patients diagnosed with IPF (please see lines 638-650 of the revised manuscript). This paragraph compiles the clinical trials on autotaxin inhibitor (GLPG190, phase 3), LPA antagonist (BMS-986020, phase 2) and the development of lipid receptors antagonist (PBI-4050, phase 2). We believe the addition made in response to this comment strengthens our therapy-focused chapter and makes the review more appealing to the readers of the Journal.

Reviewer 3 Report

The paper by Burgy C entitled "Extracellular lipids in the lung and their role in pulmonary fibrosis" was reviewed. This paper described the roles of extracellular lipids in pulmonary fibrosis and describes their roles in the deadly disorder, pulmonary fibrosis. The paper is well-written and worthy of publication in Cells. The paper is exclusively written on lipids. If the authors added a description of the interaction of lipids with extracellular matrix proteins such as osteopontin, this paper will be of interest to more readers. 

Author Response

We thank the reviewer for the positive comments. As suggested, we have emphasized the importance of ECM in the physiopathology of pulmonary fibrosis (please see lines 86-88). Further, we have added a section reviewing the body of literature showing an intimal crosstalk between lipids and ECM proteins within the lung (please see lines 137-153). We have extended on the link between osteopontin and the metabolism of P-choline and S1P. In addition, we described how lipids can influence the production of ECM components with the example of collagen production enhanced by LPA. According to the amount of additional material, we have changed the title of the first chapter which is now entitled “The physiopathology of pulmonary fibrosis”. We think these modifications and additional material have significantly improved our manuscript and hope our review on extracellular lipids in the lung and their role in pulmonary fibrosis will be of interest to the readers of Cells.

Round 2

Reviewer 2 Report

The authors have well addressed all the concerns raised by this referee, and they have made sufficient improvement to the manuscript.